# TD or not TD: Analyzing the Role of Temporal Differencing in Deep Reinforcement Learning

**Artemij Amiranashvili**
University of Freiburg

**Alexey Dosovitskiy**
Intel Labs

**Vladlen Koltun**
Intel Labs

**Thomas Brox**
University of Freiburg

## Abstract

Our understanding of reinforcement learning (RL) has been shaped by theoretical and empirical results that were obtained decades ago using tabular representations and linear function approximators. These results suggest that RL methods that use temporal differencing (TD) are superior to direct Monte Carlo estimation (MC). How do these results hold up in deep RL, which deals with perceptually complex environments and deep nonlinear models? In this paper, we re-examine the role of TD in modern deep RL, using specially designed environments that control for specific factors that affect performance, such as reward sparsity, reward delay, and the perceptual complexity of the task. When comparing TD with infinite-horizon MC, we are able to reproduce classic results in modern settings. Yet we also find that finite-horizon MC is not inferior to TD, even when rewards are sparse or delayed. This makes MC a viable alternative to TD in deep RL.

## 1 Introduction

The use of deep networks as function approximators has significantly expanded the range of problems that can be successfully tackled with reinforcement learning. However, there is little understanding of when and why certain deep RL algorithms work well. Theoretical results are mainly based on tabular environments or linear function approximators (Sutton & Barto, 2017). Their assumptions do not cover the typical application domains of deep RL, which feature extremely high input dimensionality (typically in the tens of thousands) and the use of nonlinear function approximators. Thus, our understanding of deep RL is based primarily on empirical results, and these empirical results guide the design of deep RL algorithms.

One design decision shared by the vast majority of existing value-based deep RL methods is the use of temporal difference (TD) learning – training predictive models by bootstrapping based on their own predictions. This design decision is primarily based on evidence from the pre-deep-RL era (Sutton, 1988; 1995). The results of those experimental studies are well-known and clearly demonstrate that simple supervised learning, also known as Monte Carlo prediction (MC), is outperformed by pure TD learning, which, in turn, is outperformed by TD($\lambda$) – a method that can be seen as a mixture of TD and MC (Sutton, 1988).

However, recent research has shown that an algorithm based on Monte Carlo prediction can outperform TD-based methods on complex sensorimotor control tasks in three-dimensional, partially observable environments (Dosovitskiy & Koltun, 2017). These results suggest that the classic understanding of the relative performance of TD and MC may not hold in modern settings. This evidence is not conclusive: the algorithm proposed by Dosovitskiy & Koltun (2017) involves custom components such as parametrized goals and decomposed rewards, and therefore cannot be directly compared to TD-based baselines.

In this paper, we perform a controlled experimental study aiming at better understanding the role of temporal differencing in modern deep reinforcement learning, which is characterized by essentially infinite-dimensional state spaces, extremely high observation dimensionality, partial observability, and deep nonlinear models used as function approximators. We focus on environments with visual inputs and discrete action sets, and algorithms that involve prediction of value or action-value functions. This is in contrast to value-free policy optimization algorithms (Schulman et al., 2015; Levine & Koltun, 2013) and tasks with continuous action spaces and low-dimensional vectorial state representations that have been extensively benchmarked by Duan et al. (2016) and Henderson et al.

(2017). We base our study on deep $Q$-learning (Mnih et al., 2015), where the $Q$-function is learned either via temporal differencing or via a finite-horizon Monte Carlo method. To ensure that our conclusions are not limited to pure value-based methods, we additionally evaluate asynchronous advantage actor-critic (A3C), which combines temporal differencing with a policy gradient method (Mnih et al., 2016).

Our main focus is on performing controlled experiments, in terms of both algorithm configurations and environment properties. This is in contrast to prior work, which typically benchmarked a number of existing algorithms on a set of standard environments. While proper benchmarking is crucial for tracking progress in the field, it is not always sufficient for understanding the reasons behind good or poor performance. In this work, we ensure that the algorithms are comparable by implementing them in a common software framework. By varying the parameters such as the balance between TD and MC in the learning update or the prediction horizon, we are able to clearly isolate the effect of these parameters on learning. Moreover, we designed a series of controlled scenarios that focus on specific characteristics of RL problems: reward sparsity, reward delay, perceptual complexity, and properties of terminal states. Results in these environments shed light on the strengths and weaknesses of the considered algorithms.

Our findings in modern deep RL settings both support and contradict past results on the merits of TD. On the one hand, value-based infinite-horizon methods perform best with a mixture of TD and MC; this is consistent with the TD($\lambda$) results of Sutton (1988). On the other hand, in sharp contrast to prior beliefs, we observe that Monte Carlo algorithms can perform very well on challenging RL tasks. This is made possible by simply limiting the prediction to a finite horizon. Surprisingly, finite-horizon Monte Carlo training is successful in dealing with sparse and delayed rewards, which are generally assumed to impair this class of methods. Monte Carlo training is also more stable to noisy rewards and is particularly robust to perceptual complexity and variability.

## 2 PRELIMINARIES

We work in a standard reinforcement learning setting of an agent acting in an environment over discrete time steps. At each time step $t$, the agent receives an observation $\mathbf{o}_t$ and selects an action $\mathbf{a}_t$. We assume partial observability: the observation $\mathbf{o}_t$ need not carry complete information about the environment and can be seen as a function of the environment's "true state". We assume an episodic setup, where an episode starts with time step $0$ and concludes at a terminal time step $T$. We denote by $\mathbf{s}_t$ the tuple of all observations collected by the agent from the beginning of the episode: $\mathbf{s}_t = \langle \mathbf{o}_0, \ldots, \mathbf{o}_t \rangle$. (In practice we will only include a set of recent observations in $\mathbf{s}$.) The objective is to find a policy $\pi(\mathbf{a}_t|\mathbf{s}_t)$ that maximizes the expected return – the sum of all future rewards through the remainder of the episode:

$$R_t = \sum_{i=t}^{T} r_i. \tag{1}$$

This sum can become arbitrarily large for long episodes. To avoid divergence, temporally distant rewards can be discounted. This is typically done in one of two ways: by introducing a discount factor $\gamma$ or by truncating the sum after a fixed number of steps (horizon) $\tau$.

$$R_t^{\gamma} = \sum_{i=t}^{T} \gamma^{i-t} r_i = r_t + \gamma r_{t+1} + \gamma^2 r_{t+2} + \ldots \quad ; \quad R_t^{\tau} = \sum_{i=t}^{t+\tau} r_i. \tag{2}$$

The parameters $\gamma$ and $\tau$ regulate the contribution of temporally distant rewards to the agent's objective. In what follows $\hat{R}_t$ stands for $R_t^{\gamma}$ or $R_t^{\tau}$.

For a given policy $\pi$, the value function and the action-value function are defined as expected returns that are conditioned, respectively, on the observation or the observation-action pair:

$$V^{\pi}(\mathbf{s}_t) = \mathbb{E}_{\pi}[\hat{R}_t|\mathbf{s}_t], \quad Q^{\pi}(\mathbf{s}_t, \mathbf{a}_t) = \mathbb{E}_{\pi}[\hat{R}_t|\mathbf{s}_t, \mathbf{a}_t]. \tag{3}$$

Optimal value and action-value functions are defined as the maxima over all possible policies:

$$V^{\star}(\mathbf{s}_t) = \max_{\pi} V^{\pi}(\mathbf{s}_t), \quad Q^{\star}(\mathbf{s}_t, \mathbf{a}_t) = \max_{\pi} Q^{\pi}(\mathbf{s}_t, \mathbf{a}_t). \tag{4}$$

In value-based, model-free reinforcement learning, the value or action value are estimated by a function approximator $V$ with parameters $\theta$. The function approximator is typically trained by minimizing a loss between the current estimate and a target value:

$$\mathcal{L}(\theta) = \left(V(\mathbf{s}_t; \theta) - V_{\text{target}}\right)^2.$$ (5)

The learning procedure for the action-value function is analogous. Hence, we focus on the value function in the remainder of this section.

Reinforcement learning methods differ in how the target value is obtained. The most straightforward approach is to use the empirical return as target: i.e., $V_{\text{target}} = R_t^\gamma$ or $V_{\text{target}} = R_t^\tau$. This is referred to as Monte Carlo (MC) training, since the empirical loss becomes a Monte Carlo estimate of the expected loss. Using the empirical return as target requires propagating the environment forward before a training step can take place – by $\tau$ steps for finite-horizon return $R_t^\tau$ or until the end of the episode for discounted return $R_t^\gamma$. This increases the variance of the target value for long horizons and large discount factors.

An alternative to Monte Carlo training is temporal difference (TD) learning (Sutton, 1988). The idea is to estimate the return by bootstrapping from the function approximator itself, after acting for a fixed number of steps $n$:

$$V_{\text{target}} = \sum_{i=t}^{t+n-1} \gamma^{i-t} r_i + \gamma^n V(\mathbf{s}_{t+n}; \theta).$$ (6)

TD learning is typically used with infinite-horizon returns. When the rollout length $n$ approaches infinity (or, in practice, maximal episode duration $T_{\max}$), TD becomes identical to Monte Carlo training. TD learning applied to the action-value function is known as $Q$-learning (Watkins, 1989; Watkins & Dayan, 1992; Peng & Williams, 1996; Mnih et al., 2015).

An alternative to value-based methods are policy-based methods, which directly parametrize the policy $\pi(\mathbf{a}|\mathbf{s}; \theta)$. An approximate gradient of the expected return is computed with respect to the policy parameters, and the return is maximized using gradient ascent. Williams (1992) has shown that an unbiased estimate of the gradient can be computed as $\nabla_\theta \log \pi(\mathbf{a}|\mathbf{s}; \theta) \left(R_t - b_t(\mathbf{s}_t)\right)$, where the function $b_t(\mathbf{s}_t)$ is called a baseline and can be chosen so as to decrease the variance of the estimator. A common choice for the baseline is the value function: $b_t(\mathbf{s}_t) = V^\pi(\mathbf{s}_t)$. A combination of policy gradient with a baseline value function learned via TD is referred to as an actor-critic method, with policy $\pi$ being the actor and the value function estimator being the critic.

## 3 EXPERIMENTAL SETUP

### 3.1 ALGORITHMS

In our analysis of temporal differencing we focus on three key characteristics of RL algorithms. The first is the balance between TD and MC in the learning update. The second is the prediction horizon, in particular infinite versus finite horizon. The third is the use of pure value-based learning versus an actor-critic approach which includes an explicitly parametrized policy.

To study the first aspect, we use asynchronous n-step Q-learning (n-step $Q$) (Mnih et al., 2016). In this algorithm, an action-value function is learned with n-step TD (Eq. (6)), and actions are selected greedily according to this function. By varying the rollout length $n$, we can smoothly interpolate between pure TD and pure MC updates. In order to analyze the second aspect, we implemented a finite-horizon Monte Carlo version of n-step $Q$, which we call $Q_{MC}$. This algorithm can be seen as a simplified version of Direct Future Prediction (Dosovitskiy & Koltun, 2017). Finally, we select asynchronous advantage actor-critic (A3C) (Mnih et al., 2016) to study the third aspect. In A3C, the value function estimate is learned with n-step TD, and a policy is trained with policy gradient. This allows us to evaluate the interplay of TD learning and policy gradient learning.

To ensure that the comparison is fully controlled and fair, we implemented all algorithms in the asynchronous training framework proposed by Mnih et al. (2016). Multiple actor threads are running in parallel and send the weight updates asynchronously to a parameter server. For A3C and n-step $Q$, we use the algorithms as described by Mnih et al. (2016). $Q_{\text{MC}}$ is the n-step $Q$ algorithm where the

n-step TD targets are replaced by finite-horizon MC targets. Further details on the $Q_{\mathrm{MC}}$ and n-step $Q$ algorithms and the network architecture are provided in the supplement.

Note that switching to finite horizon necessitates a small additional change in the $Q_{\mathrm{MC}}$ algorithm. In practice, in n-step $Q$ each parameter update is not just an $n$-step TD update, but a sum of all updates for rollouts from $1$ to $n$. This improves the stability of training. In $Q_{MC}$ such accumulation of updates is impossible, since predictions for different horizons are not compatible. We therefore always predict several $Q$-values corresponding to different horizons, similar to Dosovitskiy & Koltun (2017). Specifically, for horizon $\tau = 2^K$, we additionally predict $Q$-values for horizons $\{2^k\}_{0 \le k < K}$. This design choice is further explained and supported with experiments in the supplement. Apart from this, there is no difference between n-step $Q$ and $Q_{MC}$.

## 3.2 ENVIRONMENTS

To calibrate our implementations against results available in the literature, we begin by conducting experiments on several standard benchmark environments: five Atari games from the Arcade Learning Environment (Bellemare et al., 2013) and two environments based on first-person-view 3D simulation in the ViZDoom framework (Kempka et al., 2016). We used a set of Atari games commonly analyzed in the literature: Space Invaders, Pong, Beam Rider, Sea Quest, and Frostbite (Mnih et al., 2015; Schulman et al., 2015; Lake et al., 2017). For the ViZDoom environments, we used the Navigation, Battle and Battle2 scenarios from Dosovitskiy & Koltun (2017).

Our main experiments are on sequences of specialized environments. Each sequence is designed such that a single factor of variation is modified in a controlled fashion. This allows us to study the effect of this factor. Factors of variation include: reward sparsity, reward delay, reward type, and perceptual complexity.

For the controlled environments, we used the ViZDoom platform. This platform is compatible with existing map editors with built-in scripting, which allows for flexible and controlled specification of different scenarios. In comparison to Atari games, ViZDoom offers a more realistic setting with a three-dimensional environment and partially observed first-person navigation. We now briefly describe the tasks. Further details are provided in the supplement.

**Basic health gathering.** The basis for our controlled scenarios is the health gathering task. In this scenario, the agent's aim is to collect health kits while navigating through a maze using visual input. Figure 1(b) shows a typical image observed by the agent. The agent's health level is constantly declining. Health kits add to the health level. The goal is to collect as many health kits as possible. To be precise, the agent loses 6 health units every 8 steps, and obtains 20 health units when collecting a health pack. The agent's total health cannot exceed 100. The reward is $+1$ when the agent collects a health kit and 0 otherwise. There are 16 health kits in the labyrinth at any given time. When the agent collects one of them, a new one appears at a random location. An episode is terminated after 525 steps, which is equivalent to 1 minute of in-game time.

**Terminal states.** To test the effect of terminal states on the performance of the algorithms, we modified the health gathering scenario so that each episode terminates after $m$ health kits are collected. For $m = 1$, all useful training signals come from the terminal state. With larger $m$, the importance of terminal states diminishes.

**Delayed rewards.** In this sequence of scenarios we introduce a delay between the act of collecting a health kit and its effect – an increase in health and a reward of 1. We have set up environments with delays of 2, 4, 8, 16, and 32 steps.

**Sparse rewards.** To examine the effect of reward sparsity, we varied the number of available health kits on the map. We created two variations of the basic health gathering environment with increasingly sparse rewards. In the 'Sparse' setting, there are 4 health kits in the labyrinth – four times fewer than in the basic setting. In the 'Very Sparse' setting, only 2 health kits are in the labyrinth – eight times fewer than in the basic setting. In order to isolate the effect of sparsity, we keep the achievable reward fixed by adjusting the amount of health the agent loses per time period: 3 in the Sparse configuration and 2 in Very Sparse.

In the Very Sparse scenario under random exploration, the agent gathers a health kit on average every 6,440 steps.

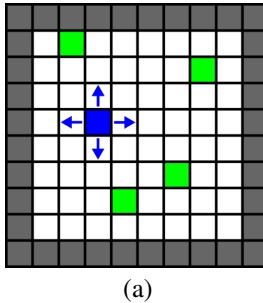 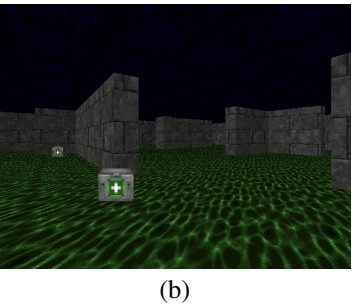 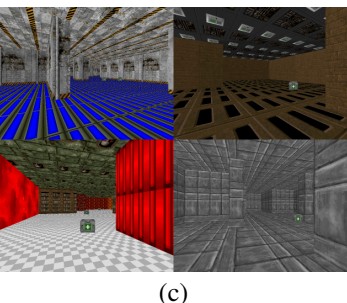

(a)                    (b)                    (c)

Figure 1: Different levels of perceptual complexity in the health gathering task. (a) Map view of a grid world. (b) First-person view of a three-dimensional environment, fixed textures. (c) First-person view of a three-dimensional environment, random textures.

**Reward type.** In this scenario, we compare the standard binary reward with its more natural but more noisy counterpart. In the basic scenario above, the reward is $+1$ for gathering a health kit and $0$ otherwise. A more natural measure of success in the health gathering task is the actual change in health. With this reward, the agent would directly aim to maximize its health. In this configuration we therefore use a scaled change in health as the reward signal. This reward is more challenging than the basic binary reward due to its noisiness (health is decreased only every eighth step) and the variance in the reward after collecting a health kit due to the total health limit.

**Perceptual complexity.** To analyze the effect of perceptual complexity, we designed variants of the health gathering task with different input representations. First, to increase the perceptual complexity of the task, we replaced the single maze used in the basic health gathering scenario by 90 randomly textured versions, some of which are shown in Figure 1(c). The labyrinth's texture is changed after each episode during both training and evaluation.

We also created two variants of the health gathering task with reduced visual complexity. These are the only controlled scenarios not using the ViZDoom framework. Both are based on a grid world, where the agent is navigating an $8\times8$ room with 5 available actions: wait, up, down, left, and right. There are 4 randomly placed health kits in the room, and the aim of the agent is to collect these, with reward $+1$ for collecting a health kit and $0$ otherwise. Each time a health kit is collected a new one appears in a random location. The two variants differ in the representation that is fed to the agent. In one, the agent's input is a 10-dimensional vector that concatenates the 2D Cartesian coordinates of the agent itself and the 4 health kits, sorted by their distance to the agent. In the other variant, we use a k-hot vector for the health kits coordinates and a one-hot vector for the agent coordinates. Each possible position on the grid is a separate entry in those vectors, and is equal to 1 if the according object is present and 0 otherwise.

### 3.3 ALGORITHM DETAILS

We used identical network architectures for the three algorithms in all experiments. For experiments in Atari and ViZDoom domains we used deep convolutional networks similar to the one used by Mnih et al. (2015). For gridworld experiments we used fully-connected networks with three hidden layers. For $Q_{\text{MC}}$ and n-step $Q$ we used dueling network architectures, similar to Wang et al. (2016). The exact architectures are specified in the supplement.

For experiments in Atari environments we followed a common practice and fed the 4 most recent frames to the networks. In all other environments the input was limited to the observation from the current time step. In ViZDoom scenarios, in addition to the observed image we fed a vector of measurements to all networks. The measurements are the agent's scalar health in the health gathering scenarios and a three-dimensional vector of the agent's health, ammo, and frags in the battle scenario.

We trained all models with 16 asynchronous actor threads, for a total of 60 million steps. We identified optimal hyperparameters for each algorithm via a hyperparameter search on a subset of environments and used these fixed hyperparameters for all environments, unless noted otherwise.

| | #steps | Atari | | | | | ViZDoom | | |
| | | Seaquest | S. Invaders | Frostbite | Pong | BeamRider | Navigat. | Battle | Battle 2 |
|---|---|---|---|---|---|---|---|---|---|
| A3C (Mnih et al., 2016) | 80M | 2300 | 2215 | 180 | 11.4 | 13236 | – | – | – |
| DFP (Dosovitskiy & Koltun, 2017) | 50M | – | – | – | – | – | 84.1 | 33.5 | 16.5 |
| $Q_{MC}$ | 60M | **12708** | 1221 | 1311 | −4.2 | 1839 | **84.4** | **35.9** | **17.5** |
| 20-step $Q$ | 60M | 4276 | 1888 | **3875** | 8.9 | **9088** | 75.7 | 32.4 | 16.0 |
| 20-step A3C | 60M | 2021 | **1952** | 202 | **20.6** | 7190 | 70.8 | 22.1 | 11.0 |

Table 1: Calibration against published results on standard environments. We report the average score at the end of an episode for Atari games, health for the Navigation scenario, and frags for the Battle scenarios. In all cases, higher is better.

For evaluation, we trained three models on each task, selected the best-performing snapshot for each training run, and averaged the performance of these three best-performing snapshots. Further details are provided in the supplement.

The implementation of the environments and a video of an $Q_{MC}$ agent trained on various tasks are available on the project page: `https://lmb.informatik.uni-freiburg.de/projects/tdornottd/`

## 4 RESULTS

### 4.1 CALIBRATION

We start by calibrating our implementations of the methods against published results reported in the literature. To this end, we train and test our implementations on standard environments used in prior work. The results are summarized in Table 1. Our implementations perform similarly to corresponding results reported in prior work.

For A3C the results are significantly different only for BeamRider. However, in Mnih et al. (2016) the evaluation used the average over the best 5 out of 50 experiments with different learning rates. We used the average over 3 runs with a fixed learning rate. Since the results for BeamRider have a high variance even for very small learning rate changes, this explains the difference between the results.

On the ViZDoom scenarios, the $Q_{MC}$ implementation performs on par with the DFP algorithm. This shows that DFP does not crucially depend on a decomposition of the reward into a vector of measurements, and can perform equally well given a standard RL setup with a scalar reward. Our A3C implementation achieves significantly better results than those reported by Dosovitskiy & Koltun (2017) on the ViZDoom scenarios. We attribute this to (a) using a rollout value of 20 in our experiments instead of 5 as used by Mnih et al. (2016) and Dosovitskiy & Koltun (2017), and (b) providing the measurements as input to the network. Dosovitskiy & Koltun (2017) did not report results on Atari games. We find that in these environments $Q_{MC}$ performs worse overall than 20-step $Q$ and 20-step A3C.

### 4.2 VARYING THE ROLLOUT IN TD-BASED ALGORITHMS

By changing the rollout length $n$ in n-step $Q$ and A3C, we can smoothly transition between TD and MC training. 1-step rollouts correspond to pure bootstrapping as used in the standard Bellman equation. Infinite rollouts (until the terminal state), on the other hand, correspond to pure Monte Carlo learning of discounted infinite-horizon returns.

Results on three environments – Basic health gathering, Sparse health gathering, and Battle – are presented in Figure 2. Rollout length of 20 is best on all tasks for n-step $Q$. Both very short and very long rollouts lead to decreased performance. These findings are in agreement with prior results of TD($\lambda$) experiments (Sutton, 1988; 1995), considering that longer rollouts increase the MC portion of the value target, converging to a full MC update for infinite rollout. A mixture of TD and MC yields the best performance. The results for A3C are qualitatively similar, and again the 20-step rollout is overall near-optimal.

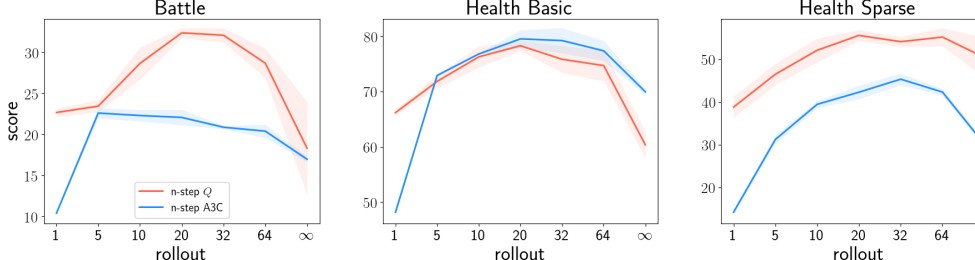

Figure 2: Effect of rollout length on TD learning for n-step $Q$ and A3C. We report average health at the end of an episode for health gathering and average frags in the Battle scenario. Higher is better.

### 4.3 Controlled Experiments

We now proceed to a series of controlled experiments on a set of specifically designed environments and compare TD-based methods to $Q_{MC}$, a purely Monte Carlo approach. The motivation is as follows. In the previous section we have seen that very long rollouts lead to deteriorated performance of n-step $Q$ and A3C. This can be attributed to large variance in target values. The variance can be reduced by using a finite horizon, as is the case in $Q_{MC}$. However, the use of a finite horizon means that rewards that are further away than the horizon will not be part of the value target, resulting in a disadvantage in tasks with sparse or delayed rewards. In order to evaluate this we run controlled experiments designed to isolate the reward delay, sparsity, and other factors. We test 20-step $Q$ and A3C (optimal rollout for TD-based methods), 5-step $Q$ and A3C (more TD in the update), and $Q_{MC}$ (finite horizon Monte Carlo).

**Reward type.** We contrast the standard binary reward with the more natural reward signal proportional to the change in the health level of the agent. Figure 3 (left) shows that in the scenario with binary reward the performance of $Q_{MC}$, 20-step $Q$, and 20-step A3C is nearly identical, within $4\%$ of each other. However, when trained with the noisier health-based reward, $Q_{MC}$ performs within $1\%$ of the result with binary reward, but the performance of TD-based algorithms decreases significantly, especially for the 5-step rollouts. These results suggest that Monte Carlo training is more robust to noisy rewards than TD-based methods.

**Terminal states.** Table 2 shows that in environments where terminal states play a crucial role, $Q_{MC}$ is outperformed by TD-based methods. This is due to the finite-horizon nature of $Q_{MC}$. A terminal reward only contributes to a single update per episode, while in TD it contributes to every update in the episode. If non-terminal rewards are present ($m = 2$), $Q_{MC}$ approaches the TD-based algorithms, but still does not reach the performance of 20-step Q. Difficulties with terminal states can partially explain poor performance of $Q_{MC}$ on some Atari games. The results for larger $m$ values are discussed in the supplement.

|  | $m = 1$ | $m = 2$ |
|---|---|---|
| $Q_{MC}$ | 43.3 | 64.0 |
| 20-step $Q$ | **75.9** | **75.5** |
| 5-step $Q$ | 74.3 | 71.3 |
| 20-step A3C | 64.7 | 58.2 |
| 5-step A3C | 61.1 | 52.3 |

Table 2: Terminal states.

**Delayed rewards.** Figure 3 (middle) shows that the performance of all algorithms declines even with moderate delays in the reward signal. A delay of 2 steps, or approximately 0.2 seconds of in-game time, already leads to a 8–12% relative drop in performance for $Q_{MC}$ and 20-step TD algorithms and a 30–40% drop for 5-step TD algorithms. With a delay of 8 steps, or approximately 1 second, the performance of $Q_{MC}$ and 20-step TD algorithms drops by 30–70% and 5-step TD agents are essentially unable to survive until the end of an episode. With a delay of 32 steps, all algorithms degrade to a trivial score. Interestingly, the performance of $Q_{MC}$ declines less rapidly than the performance of the other algorithms and $Q_{MC}$ consistently outperforms the other algorithms in the presence of delayed rewards.

**Sparse rewards.** TD-based infinite-horizon approaches should theoretically be effective at propagating distal rewards, and are therefore supposed to be advantageous in scenarios with sparse rewards. The results on the Sparse and Very Sparse scenarios however, do not support this expectation

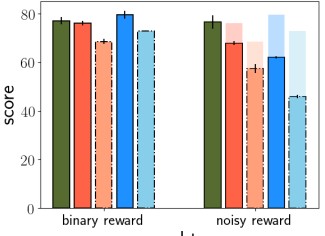 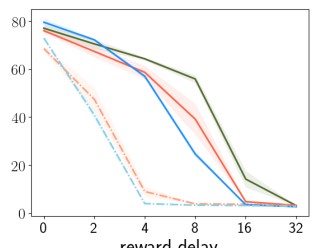 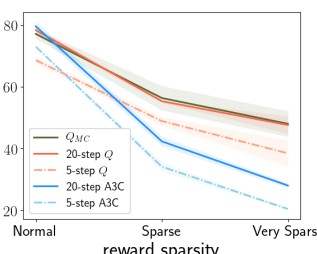

Figure 3: Effect of reward properties. Left to right: reward type, reward delay, reward sparsity. We report the average health at the end of an episode. Higher is better. MC training ($Q_{MC}$, green) performs well on all environments.

(Figure 3 (right)): $Q_{MC}$ performs on par with 20-step $Q$, and noticeably better than 20-step A3C and 5-step algorithms. We believe the reason for the unexpectedly good performance of $Q_{MC}$ is that Monte Carlo approaches are well suited for training perception systems, as discussed in more detail in Section 4.4.

**Perceptual complexity.** We test the algorithms on a series of environments with varying perceptual complexity. The results are summarized in Figure 4. In gridworld environments, TD-based methods perform well. The Coord. Grid task, where the task is simplified by sorting the health kit coordinates by distance, is successfully solved by all methods. 5-step unrolling outperforms the 20-step versions and $Q_{MC}$ in both setups.

However, the situation is completely different in the vision-based Basic and Multi-texture setups, in which the perceptual input is much more complex. In the Basic setup, all methods perform roughly on par, but 5-step unrolling drops behind the other methods. In the Multi-texture setup, $Q_{MC}$ outperforms other algorithms.

To further analyze the effect of perception on DRL, we conduct an additional experiment where we separate the learning of perception and control. We first train two perception systems on the Battle task by predicting $Q$-values under a fixed policy with 20-step $Q$ or $Q_{MC}$. We then re-initiailize the weights in the top two layers, freeze the weights in the rest of the the networks, and re-train the top two layers on the Battle task with 20-step $Q$ or $Q_{MC}$. To make sure that the perception results are not the result of having multiple heads for multiple final horizons, we also trained one perception using a single head (1-head $Q_{MC}$). Further details are provided in the supplement. The results are shown in Table 3. Both $Q$ and $Q_{MC}$ control reach higher score with a perception system trained with $Q_{MC}$. This supports the hypothesis that Monte Carlo training is efficient at training deep perception systems from raw pixels.

|  | Control | |
|---|---|---|
| Perception | 20-step $Q$ | $Q_{MC}$ |
| 20-step $Q$ | 18.0 | 19.9 |
| $Q_{MC}$ | **31.8** | **35.2** |
| 1-head $Q_{MC}$ | 30.8 | 30.6 |

Table 3: Separate training of perception and control on the Battle scenario. Higher is better.

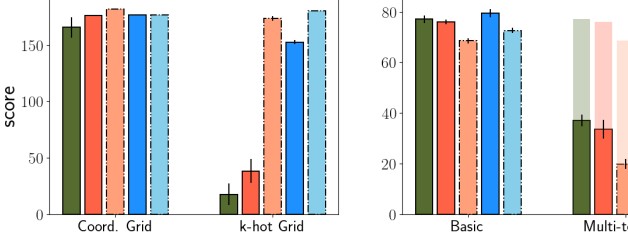

Figure 4: Effect of perceptual complexity. We report average cumulative reward per episode for grid worlds and average health at the end of the episode for ViZDoom-based setups. Perception in both gridworlds is trivial. The perceptual complexity in the multi-texture task is higher than in the basic task.

### 4.4 TD OR NOT TD?

Temporal differencing methods are generally considered superior to Monte Carlo methods in reinforcement learning. This opinion is largely based on empirical evidence from domains such as gridworlds (Sutton, 1995), cart pole (Barto et al., 1983), and mountain car (Moore, 1990). Our results agree: in gridworlds and on Atari games we find that n-step $Q$ learning outperforms $Q_{MC}$. We further find, similar to the TD($\lambda$) experiments from the past (Sutton, 1988), that a mixture of MC and TD achieves best results in n-step $Q$ and A3C.

However, the situation changes in perceptually complex environments. In our experiments in immersive three-dimensional simulations, a finite-horizon MC method ($Q_{MC}$) matches or outperforms TD-based methods. Especially interesting are the results of the sparse reward experiments. Sparse problems are supposed to be specifically challenging for finite-horizon Monte Carlo estimation: in our Very Sparse setting, average time between health kits is $44$ time steps when a human is controlling the agent. This exceeds $Q_{MC}$'s finite prediction horizon of 32 steps, making it seemingly impossible for the algorithm to achieve nontrivial performance. Yet $Q_{MC}$ is able to keep up with the results of the 20-step $Q$ algorithm and clearly outperforms A3C.

What is the reason for this contrast between classic findings and our results? We believe that the key difference is in the complexity of perception in immersive three-dimensional environments, which was not present in gridworlds and other classic problems, and is only partially present in Atari games. In immersive simulation, the agent's observation is a high-dimensional image that represents a partial view of a large (mostly hidden) three-dimensional environment. The dimensionality of the state space is essentially infinite: the underlying environment is specified by continuous surfaces in three-dimensional space. Memorizing all possible states is easy and routine in gridworlds and is also possible in some Atari games (Blundell et al., 2016), but is not feasible in immersive three-dimensional simulations. Therefore, in order to successfully operate in such simulations, the agent has to learn to extract useful representations from the observations it receives. Encoding a meaningful representation from rich perceptual input is where Monte Carlo methods are at an advantage due to the reliability of their training signals. Monte Carlo methods train on ground-truth targets, not "guess from a guess", as TD methods do (Sutton & Barto, 2017).

These intuitions are supported by our experiments. Figure 4 shows that increasing the perceptual difficulty of the health gathering scenario hurts the performance of $Q_{MC}$ less than it does the TD-based approaches. Table 3 shows that $Q_{MC}$ is able to learn a better perception network than 20-step $Q$. In Figure 3, 20-step TD algorithms perform better than their 5-step counterparts in all tested scenarios. Longer rollouts bring TD closer to MC, in agreement with our hypothesis.

## 5 CONCLUSION

For the past 30 years, TD methods have dominated the field of reinforcement learning. Our experiments on a range of complex tasks in perceptually challenging environments show that in deep reinforcement learning, finite-horizon MC can be a viable alternative to TD. We find that while TD is at an advantage in tasks with simple perception, long planning horizons, or terminal rewards, MC training is more robust to noisy rewards, effective for training perception systems from raw sensory inputs, and surprisingly successful in dealing with sparse and delayed rewards. A key challenge motivated by our results is to find ways to combine the advantages of supervised MC learning with those of TD. We hope that our work will contribute to a set of best practices for deep reinforcement learning that are consistent with the empirical reality of modern application domains.

### ACKNOWLEDGMENTS

This project was funded in part by the BrainLinks-BrainTools Cluster of Excellence (DFG EXC 1086) and by the Intel Network on Intelligent Systems.

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

## Supplementary Material

### S1    Further results

**Effect of the rollout length and the prediction horizon**    In Table 5 of the main paper we have shown that the performance of n-step $Q$ decreases for rollouts larger than 20. For the $Q_{\text{MC}}$ algorithm a similar phenomenon is observed for large horizons, as shown in Table S1. The performance is decreasing for a horizon larger than 32.

In both cases, the decrease is likely caused by the high variance of large sums of future rewards. The high variance in reward sums increases the variance of the gradients and leads to higher noise when training the value predictions. This hinders the action selection process, which relies on fine differences between values of different actions.

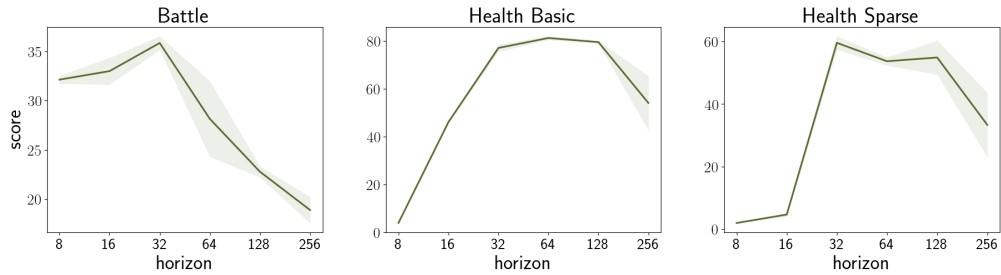

Figure S1: Performance of the $Q_{\text{MC}}$ algorithm using different value prediction horizons.

**Difference between asynchronous n-step $Q$ and $Q_{\text{MC}}$.**    As mentioned in the main paper, apart from the different targets to learn the $Q$-function there is another difference between the n-step $Q$ and $Q_{\text{MC}}$ algorithms. It is caused by the usage of multiple unrolling values in the n-step $Q$ algorithms. In n-step $Q$ instead of only using the n-step rollout, multiple values are used within every batch (every value from 1 to n (Mnih et al., 2016)). This results in an increased performance and stability of the n-step $Q$ algorithm. It is not directly applicable to $Q_{\text{MC}}$ since different unrolling values result in different finite horizons. Instead in $Q_{\text{MC}}$ multiple $Q$-function heads exist to predict the different finite horizons (Dosovitskiy & Koltun, 2017). The difference between the trivial implementation and the multiple unrolling modifications are shown in Table S1.

There are no further differences between the two algorithms. They use the same architecture and asynchronous training. Both even perform best under the same hyperparameters like the learning rate.

|  | Health Basic | Health Sparse | Battle |
|---|---|---|---|
| $Q_{\text{MC}}$ | 77.1 | 59.5 | 35.9 |
| Constant rollout $Q_{\text{MC}}$ | 78.2 | 55.9 | 30.9 |
| 20-step $Q$ | 78.3 | 55.3 | 32.4 |
| Constant rollout 20-step $Q$ | 69.0 | 45.5 | 23.5 |

Table S1: Difference between using multiple or constant rollouts within one train step.

**Separate training of perception and control**    In order to perform the perception freezing experiments we first train two perception systems on the Battle task with 20-step $Q$ and $Q_{MC}$ for 20 million steps (1/3 of the usual training) by predicting $Q$ values under a fixed policy (we tried using a fully trained $Q_{MC}$ or 20-step $Q$ policy). Thereafter we freeze the perception and the measurements part of the network. (The full architecture of the perception and the measurements are shown in Table S4). We then reinitialize the remaining layers and retrain the networks with the frozen perception with $Q_{MC}$ and 20-step $Q$, both using each of the two available perceptions (for 40 million steps).

|  | Pretrained using $Q_{\mathrm{MC}}$ policy | | Pretrained using 20-step $Q$ policy | |
|---|---|---|---|---|
|  | 20-step $Q$ | $Q_{\mathrm{MC}}$ | 20-step $Q$ | $Q_{\mathrm{MC}}$ |
| 20-step $Q$ perception | 18.0 | 19.9 | 16.2 | 20.3 |
| $Q_{\mathrm{MC}}$ perception | **31.8** | **35.2** | **30.6** | **30.4** |

Table S2: Performance of 20-step $Q$ and $Q_{\mathrm{MC}}$ with a pretrained and frozen perception, higher is better.

The full results are shown in Table S2. Both $Q$ and $Q_{\mathrm{MC}}$ are able to reach higher score with a $Q_{\mathrm{MC}}$ perception, on both of the used initial policies. The results in the main paper correspond to perception systems trained under the $Q_{MC}$ policy.

**Additional results on terminal states**  The full results on the terminal reward environment are shown in Table S3. As $m$ increases the terminal rewards become less relevant and for $m = \infty$ the task converges to the Health Sparse environment. Beside the result that $Q_{\mathrm{MC}}$ performs worse than the other algorithms for $m = 1$ we also see that the performance of all TD-based algorithms declines with larger $m$ values ($Q_{\mathrm{MC}}$ performance also declines after $m = 3$). The reason for this is that apart from the terminal state the task becomes harder for larger $m$ values: For $m = 1$ it is easy to find a single health kit. The larger $m$ becomes, the higher is the probability that a new health kit will spawn in a hard to reach place. Overall, exploring the labyrinth efficiently is important for high scores on the Health Sparse ($m = \infty$) task, but complete labyrinth exploration is not needed to find a small amount of health kits.

To show this we evaluated the performance of an 20-step $Q$ agent, trained on the Health Sparse environment, on the $m = 1$ task. As expected, without additional training the agent was able to solve the task with the same score $(75, 9)$ as the agent trained on the $m = 1$ task. Since the increasing difficulty is not directly related to terminal states, we excluded the results for $m > 2$ from the main paper.

|  | $m = 1$ | $m = 2$ | $m = 3$ | $m = \infty$ (Health Sparse) |
|---|---|---|---|---|
| $Q_{\mathrm{MC}}$ | 43.3 | 64.0 | 64.1 | **56.3** |
| 20-step $Q$ | **75.9** | **75.5** | **71.8** | 55.3 |
| 5-step $Q$ | 74.3 | 71.3 | 67.5 | 48.9 |
| 20-step A3C | 64.7 | 58.2 | 53.7 | 42.3 |
| 5-step A3C | 61.1 | 52.3 | 45.7 | 34.1 |

Table S3: Difference between using multiple or constant rollouts within one train step.

## S2 Additional algorithm and environment details

**Batch size for small rollouts**  In algorithms with asynchronous n-step TD targets the batch size is usually equal to the unrolling length n. However decreasing the batch size in A3C could also effect the performance of the policy gradient of the A3C loss. To make sure that we only measure the effect of different n-step TD targets and do not alternate the policy gradient part we keep the batch size at the constant value of 20 (for all rollouts smaller than 20). This is realized by using multiple n-step rollouts within one batch (e.g. for a 5-step rollout the batch consists of 4 rollouts). Overall those batches lead to improved performance of A3C. For n-step $Q$ using the constant batch size of 20 results in similar performance and significantly reduced the execution time. Therefore we used those batches for both A3C and n-step $Q$ in our experiments.

**$Q_{\mathrm{MC}}$ and n-step $Q$ details**  In each experiment we used the same network architecture for all algorithms. For tasks with visual input – in ViZDoom and ALE – we used a convolutional network with architecture similar to Mnih et al. (2015). For all experiments in the ViZDoom environment, in addition to the image the networks got a vector of measurements as input: agent's health level

and current time step for Health gathering and Navigation, and agent's health, ammo and frags for Battle. For $Q_{\text{MC}}$ and n-step $Q$ we used the dueling architecture (Wang et al., 2016), splitting value prediction into an action independent expectation $E(\mathbf{s}_t, \theta)$ and an action dependent part for the advantage of using a specific action $A(\mathbf{s}_t, \mathbf{a}, \theta)$. For $l$ actions, the value prediction emitted by the network is computed as:

$$Q(\mathbf{s}_t, \mathbf{a}, \theta) = E(\mathbf{s}_t, \theta) + \overline{A}(\mathbf{s}_t, \mathbf{a}, \theta) \quad ; \quad \overline{A}(\mathbf{s}_t, a, \theta) = A(\mathbf{s}_t, \mathbf{a}, \theta) - \frac{1}{l} \sum_{\mathbf{a}'} A(\mathbf{s}_t, \mathbf{a}', \theta) \quad (7)$$

The architecture of the $Q_{\text{MC}}$ Network is shown in Table S4. $Q_{\text{MC}}$ is predicting the $Q$ value for multiple finite horizons at once: 1, 2, 4, 8, 16 and 32. Predictions for all horizons are emitted at once. Therefore, for $l$ actions, the network has 6 outputs for the expectation values and $6 \times l$ outputs for the action advantages. We used greedy action selection according to an objective function which is a linear combination of predictions at different horizons, same as in DFP (Dosovitskiy & Koltun, 2017):

$$a(\mathbf{s}_t) = \arg\max_{a'} \left[ 0.5 \cdot Q^{(8)}(\mathbf{s}_t, a') + 0.5 \cdot Q^{(16)}(\mathbf{s}_t, a') + 1.0 \cdot Q^{(32)}(\mathbf{s}_t, a') \right] \quad (8)$$

The network for n-step $Q$ was identical, except that instead of 6 predictions, a single value function was predicted for each action. The A3C architecture was also identical, except that the network was not split in the last hidden layer like it was for the dueling networks. Both the policy and the value output shared the same last hidden layer as in Mnih et al. (2016). The network we are using for A3C is larger than that used by Mnih et al. (2016). We found that the larger network matches or exceeds the performance of the smaller network used by Mnih et al. (2016) on our tasks.

For both gridworlds we used fully connected networks. For all results reported in the paper the three algorithms used three hidden fully connected layers with size of 512.

**Training and evaluation details** We found that for each of the three asynchronous algorithms the learning rate of $7 \times 10^{-4}$ leads to the best result in most of the tested environments. Further we found that, in general, ViZdoom scenarios are less sensitive to learning rate changes than different Atari games. We decreased the learning rate linearly to zero over the training procedure. As the optimizer we used shared RMSProp with the same parameters as in Mnih et al. (2016).

In all experiments we used a total of 60 million steps for training. This means all actor threads together processed 60 million steps. We used frame skip of 4, therefore 60 million frame-skipped steps correspond to 240 million non-frame-skipped environment steps. For $Q_{\text{MC}}$ each asynchronous agent performed a parameter update every 20 steps with a batch size of 20. Each time the most recent 20 frames with available value targets were used. Every 2.5 million steps we evaluated the network over 500 episodes for Vizdoom Environments and over 200 episodes for Atari games. For one training run the best result out of all 24 evaluations was considered as its final score. For each experiment three runs were performed for each algorithm. The average of the three run scores was considered as the final performance of that algorithm on the task.

**Additional environment details** The Navigation scenario is identical to the "Health Gathering Supreme" scenario included in the ViZDoom environment. The aim of the agent is to navigate a maze, collect health kits and avoid vials with poison. A map of the maze is shown in Figure S2.

All other Health gathering scenarios are set up in the same labyrinth, but differ in the presence and the number of objects in the maze: no poison vials, and a different number of health kits depending on the variant of the Health gathering scenario. In each Health gathering scenario a constant number of health kits is present on the map at any given point in time. Once a health kit is gathered, another one is created at a random location in the maze.

To make sparse health gathering map results comparable to each other we kept the health $d$ that the agent looses every 8 time steps to be proportional to the density of health kits on the map: $d \propto \sqrt{\#\text{health\_kits}}$.

In the Battle scenario we used the same reward as in Dosovitskiy & Koltun (2017). It is a weighted sum of changes in measurements: $r = f + \Delta h/60 + \Delta a/20$ where $f$ are the amount of eliminated

| Network part | Input type | Input size | Channels | Kernel | Stride | Layer type |
|---|---|---|---|---|---|---|
| Perception (P) | image | $84 \times 84 \times \{1 \text{ or } 4\}$ | 32 | $8 \times 8$ | 4 | convolutions |
| | | $20 \times 20 \times 32$ | 64 | $4 \times 4$ | 2 | |
| | | $9 \times 9 \times 64$ | 64 | $3 \times 3$ | 1 | |
| | | $7 \times 7 \times 64$ | 3136 | | | flatting |
| | | 3136 | 512 | | | fully connected |
| Measurements (M) | vector | $\{2 \text{ or } 3\}$ | 128 | | | fully connected |
| | | 128 | 128 | | | |
| | | 128 | 128 | | | |
| Expectation | P + M | $512 + 128$ | 512 | | | fully connected |
| | | 512 | 6 | | | |
| Action advantage | P + M | $512 + 128$ | 512 | | | fully connected |
| | | 512 | $6 \cdot l$ | | | |

Table S4: Network architecture of $Q_{\mathrm{MC}}$ for $l$ actions.

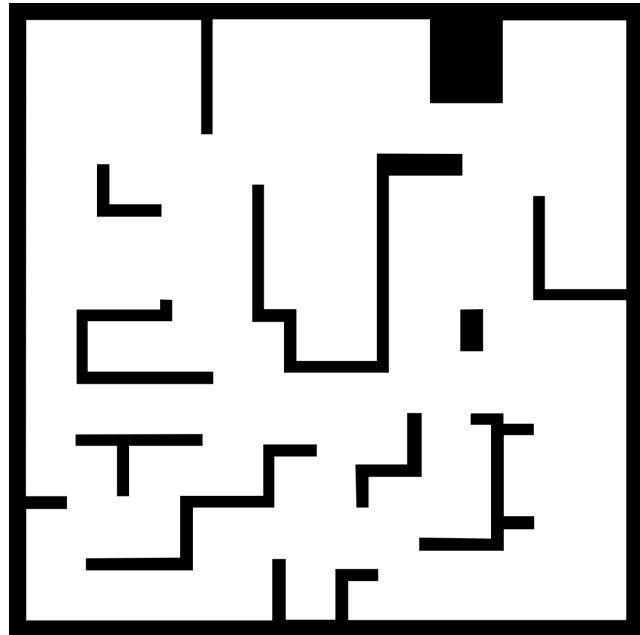

Figure S2: Map of the health gathering labyrinth.

monsters, $\Delta h$ the change in health and $\Delta a$ the change in ammunition. For Basic health gathering we either used a binary reward $r \in \{0, 1\}$, or the change in health: $r = \Delta h / 30$.

