# OpenReview forum: "TD or not TD: Analyzing the Role of Temporal Differencing in Deep Reinforcement Learning"
_ICLR.cc/2018/Conference — Accept (Poster)_

### Official Review · AnonReviewer3 · 2017-11-26
**This paper includes several controlled empirical studies comparing MC and TD methods in predicting of value function with complex DNN function approximators.**

**Rating:** 7
**Confidence:** 4

**Review:**

This paper includes several controlled empirical studies comparing MC and TD methods in predicting of value function with complex DNN function approximators. Such comparison has been carried out both in theory and practice for simple low dimensional environments with linear (and RKHS) value function approximation showing how TD methods can have much better sample complexity and overall performance compared to pure MC methods. This paper shows some results to the contrary when applying RL to complex perceptual observation space.

The main results include:
(1) In a rollout update a mix of MC and TD update (i.e. a rollout of > 1 and < horizon) outperforms either extreme. This is inline with TD-lambda analysis in previous work.
(2) Pure MC methods can outperform TD methods when the rewards becomes noisy.
(3) TD methods can outperform pure MC methods when the return is mostly dominated by the reward in the terminal state.
(4) MC methods tend to degrade less when the reward signal is delayed.
(5) Somewhat surprising: MC methods seems to be on-par with TD methods when the reward is sparse and even longer than the rollout horizon.
(6) MC methods can outperform TD methods with more complex and high dimensional perceptual inputs.

The authors conjecture that several of the above observations can be explained by the fact that the training target in MC methods is "ground truth" and do not rely on bootstrapping from the current estimates as is done in a TD rollout. They suggest that training on such signal can be beneficial when training deep models on complex perceptual input spaces.

The contributions of the paper are in parts surprising and overall interesting. I believe there are far more caveats in this analysis than what is suggested in the paper and the authors should avoid over-generalizing the results based on a few domains and the analysis of a small set of algorithms. Nonetheless I find the results interesting to the RL community and a starting point to further analysis of the MC methods (or adaptations of TD methods) that work better with image observation spaces. Publishing the code, as the authors mentioned, would certainly help with that.

Notes:
- I find the description of the Q_MC method presented in the paper very confusing and had to consult the reference to understand the details. Adding a couple of equations on this would improve the readability of the paper.

- The first mention of partial observability can be moved to the introduction.

- Adding results for m=3 to table 2 would bring further insight to the comparison.

- The results for the perceptual complexity experiment seem contradictory and inconclusive. One would expect Q_MC to work well in Grid Map domain if the conjecture put forth by the authors was to hold universally.

- In the study on reward sparsity, although a prediction horizon of 32 is less than the average steps needed to get to a rewarding state, a blind random walk might be enough to take the RL agent to a close-enough neighbourhood from which a greedy MC-based policy has a direct path to the goal. What is missing from this picture is when a blind walk cannot reach such a state, e.g. when a narrow corridor is present in the environment. Such a case cannot be resolved by a short horizon MC method. In other words, a sparse reward setting is only "difficult" if getting into a good neighbourhood requires long term planning and cannot be resolved by a (pseudo) blind random walk.

- The extrapolation of the value function approximator can also contribute to why the limited horizon MC method can see beyond its horizon in a sparse reward setting. That is, even if there is no way to reach a reward state in 32 steps, an MC value function approximation with horizon 32 can extrapolate from similar looking observed states that have a short path to a rewarding state, enough to be better than a blind random walk. It would have been nice to experiment with increasing model complexity to study such effect.

---

> ### Author Response · Authors · 2017-12-20
> **Response to Reviewer 3**
>
> We thank the reviewer for the valuable comments and detailed suggestions.
>
> > I find the description of the Q_MC method presented in the paper very confusing and had to consult the reference to understand the details. Adding a couple of equations on this would improve the readability of the paper.
>
> The equations describing the Q_MC method were in the "Network details" section of the supplement material. We have renamed the section into "Q_MC and n-step Q details"  and refer to the section in the main paper to make it easier to find this information.
>
> > The first mention of partial observability can be moved to the introduction.
>
> We changed the introduction accordingly.
>
> > Adding results for m=3 to table 2 would bring further insight to the comparison.
>
> m=3 results were added to the supplement.
>
> > The results for the perceptual complexity experiment seem contradictory and inconclusive. One would expect Q_MC to work well in Grid Map domain if the conjecture put forth by the authors was to hold universally.
>
> We think that the grid worlds were presented in the paper in a misleading manner. The two grid worlds are not clearly different in their perceptual difficulty. In one of them the agent receives its location and locations of health kits as 2D vectors of coordinates (sorted by distance to the agent) and in the other one as k-hot vectors. In both cases, the relevant information is readily available. It is not obvious which of these representations is easier for a deep network to process. We have changed the grid world names to "Coord. Grid" and "k-hot Grid", modified the Figure 4 caption, and adjusted the paper text to clarify the grid tasks results.
>
> > What is missing from this picture is when a blind walk cannot reach such a state, e.g. when a narrow corridor is present in the environment.
>
> We agree that this is an interesting problem. However, if a random agent is never reaching a reward, both TD and MC cannot improve the policy. Both rely on receiving a reward for the Q_target to start improving. We believe this problem is more related to improving on the epsilon-greedy exploration or introducing auxiliary rewards encouraging exploration, and less related to the comparison between TD and MC algorithms. One example of such a problem is the Pitfall Atari game where a random agent is unable to reach any positive rewards. To the best of our knowledge, so far no epsilon-greedy-based algorithm was able to reach a positive average reward, as for example seen for multiple algorithms in Figure 14 in Bellemare et al. "A Distributional Perspective on Reinforcement Learning", 2017.

---

### Official Review · AnonReviewer1 · 2017-11-26
**Important analysis of the role of rollout length and value function bootstrapping in RL**

**Rating:** 7
**Confidence:** 4

**Review:**

This paper revisits a subject that I have not seen revisited empirically since the 90s: the relative performance of TD and Monte-Carlo style methods under different values for the rollout length. Furthermore, the paper performs controlled experiments using the VizDoom environment to investigate the effect of a number of other environment characteristics, such as reward sparsity or perceptual complexity. The most interesting and surprising result is that finite-horizon Monte Carlo performs competitively in most tasks (with the exception of problems where terminal states play a big role (it does not do well at all on Pong!), and simple gridworld-type representations), and outperforms TD approaches in many of the more interesting settings. There is a really interesting experiment performed that suggests that this is the case due to finite-horizon MC having an easier time with learning perceptual representations. They also show, as a side result, that the reward decomposition in Dosvitskiy & Koltun (oral presentation at ICLR 2017) is not necessary for learning a good policy in VizDoom.

Overall, I find the paper important for furthering the understanding of fundamental RL algorithms. However, my main concern is regarding a confounding factor that may have influenced the results: Q_MC uses a multi-headed model, trained on different horizon lengths, whereas the other models seem to have a single prediction head. May this helped Q_MC have better perceptual capabilities?

A couple of other questions:
- I couldn't find any mention of eligibility traces - why?
- Why was the async RL framework used? It would be nice to have a discussion on whether this choice may have affected the results.

---

> ### Author Response · Authors · 2017-12-20
> **Response to Reviewer 1**
>
> We thank the reviewer for the careful review and useful suggestions.
>
> > Q_MC uses a multi-headed model, trained on different horizon lengths, whereas the other models seem to have a single prediction head. May this helped Q_MC have better perceptual capabilities?
>
> As we discuss in the supplement section "Difference between asynchronous n-step Q and Q_MC", n-step Q learning relies on the use of multiple rollouts within one batch. The only way to use multiple rollouts in a finite horizon MC setting is by having multiple heads. Therefore we used the better variants of both algorithms throughout the paper. To make sure that the multiple heads are not the reason for the better perceptual capabilities, we added perception learned by the 1-head Q_MC algorithm to the Table 3, receiving similar results to the multi head perception.
>
> > I couldn't find any mention of eligibility traces - why?
>
> Eligibility traces are used in order to implement the TD(lambda) algorithm in the backward view. In our paper we interpolate between MC and TD using n-step Q learning with different n values instead of TD(lambda). There are multiple reasons for this decision. First, most of the recent RL algorithms use a forward view implementation. Second, recent methods achieve best performance using RMS or Adam optimizers, but algorithms with eligibility traces are based on stochastic gradient descent and do not trivially carry over to other optimizers. Third, van Seijen has shown that standard TD(lambda) with eligibility traces does not perform well when combined with non-linear function approximation (van Seijen, Effective Multi-step Temporal-Difference Learning for Non-Linear Function Approximation 2016). Therefore, we decided to focus on algorithms that do not use eligibility traces.
>
> > Why was the async RL framework used?
>
> Q_MC and n-step Q (n>1) are both on-policy algorithms. We follow the asynchronous training framework for on-policy RL algorithms from Mnih at al. 2016.  Asynchronous training allows us to run experiments efficiently on CPUs, which are more easily available than GPUs. Moreover, we have found that asynchronous Q_MC achieves similar performance as synchronous DFP, therefore we expect no major impact of this implementation detail on the results.

---

### Official Review · AnonReviewer2 · 2017-11-27
**A thoughtful and well executed investigation into deep-RL techniques and challenges for robust approaches, including the relative benefits of TD versus MC.**

**Rating:** 7
**Confidence:** 4

**Review:**

The authors present a testing framework for deep RL methods in which difficulty can be controlled along a number of dimensions, including: reward delay, reward sparsity, episode length with terminating rewards, binary vs real rewards and perceptual complexity. The authors then experiment with a variety of TD and MC based deep learners to explore which methods are most robust to increases in difficulty along these dimensions. The key finding is that MC appears to be more robust than TD in a number of ways, and in particular the authors link this to domains with greater perceptual challenges.

This is a well motivated and explained paper, in which a research agenda is clearly defined and evaluated carefully with the results reflected on thoughtfully and with intuition. The authors discover some interesting characteristics of MC based Deep-RL which may influence future work in this area, and dig down a little to uncover the principles a little. The testing framework will be made public too, which adds to the value of this paper. I recommend the paper for acceptance and expect it will garner interest from the community.

Detailed comments
  • [p4, basic health gathering task] "The goal is to survive and maintain as much health
as possible by collecting health kits...The reward is +1 when the agent collects a health kit and 0 otherwise." The reward suggests that the goal is to collect as many health kits as possible, for which surviving and maintaining health are secondary.
  • [p4, Delayed rewards] It might be interesting to have a delay sampled from a distribution with some known mean. Otherwise, the structure of the environment might support learning even when the reward delay would otherwise not.
  • [p4, Sparse rewards] I am not sure it is fair to say that the general difficulty is kept fixed. Rather, the average achievable reward for an oracle (that knows whether health packs are) is fixed.
  • [p6] "Dosovitskiy & Koltun (2017) have not tested DFP on Atari games." Probably fairer/safer to say: did not report results on Atari games.

---

> ### Author Response · Authors · 2017-12-20
> **Response to Reviewer 2**
>
> We thank the reviewer for the detailed review and positive feedback.
> We have adjusted the paper according to the comments.
>
> > It might be interesting to have a delay sampled from a distribution with some known mean.
>
> We performed additional experiments with a uniformly sampled delay in the intervals of [2,6] and [6,10]. The results were nearly identical to the according experiments of 4 and 8 step delay.

---

### Decision · Program_Chairs · 2018-01-29
**ICLR 2018 Conference Acceptance Decision**

**Decision:**

Accept (Poster)

**Comment:**

This is an interesting piece of work that provides solid evidence on the topic of bootstrapping in deep reinforcement learning.